# Clinical perspectives on the identification of neurodevelopmental conditions in children and changes in referral pathways: qualitative interviews

Barry Coughlan ![ID],[1] Matt Woolgar,[2] Alissa Mann ![ID],[3] Robbie Duschinsky ![ID] [1]

[1]Department of Public Health and Primary Care, University of Cambridge, Cambridge, UK
[2]Institute of Psychiatry Psychology and Neuroscience, King's College London, London, UK
[3]Department of Psychology, University of Bath, Bath, UK

**Correspondence to**
Barry Coughlan;
bc471@medschl.cam.ac.uk

## ABSTRACT

**Objective** Previous work has raised questions about the role of general practitioners (GPs) in the identification of neurodevelopmental conditions such as autism spectrum disorders (autism) and attention deficit hyperactivity disorders (ADHD). This study aimed to explore how GPs identify these conditions in practice and their perspectives on recent changes to local referral pathways that mean referrals to the neurodevelopmental team come through educational professionals and health visitors, rather than GPs. This study also aimed to explore Child and Adolescent Mental Health Services (CAMHS) specialist's perspectives on the role of GPs.

**Setting** GP practices, local neurodevelopmental services and specialist CAMHS services in the UK. Participants: semistructured interviews were conducted with GPs (n=8), specialists in local CAMHS (n=7), and professionals at national CAMHS services around the country (n=10). Interviews were conducted between January and May 2019. A framework approach informed by thematic analysis was used to analyse the data.

**Results** GPs drew on various forms of tacit and explicit information including behavioural markers, parental report, prior knowledge of the family, expert and lay resources. Opinions varied between GPs regarding changes to the referral pathway, with some accepting the changes and others describing it as a 'disaster'. CAMHS specialists tended to feel that GPs required more neurodevelopmental training and time to conduct consultations.

**Conclusion** This study adds to the literature showing that GPs use an array of information sources when making referral decisions for autism and ADHD. Further work is urgently required to evaluate the impact of reconfiguring neurodevelopmental referral pathways such that GPs have a diminished role in identification.

## INTRODUCTION

In the UK, general practitioners (hereafter GPs) are one of the main providers of primary healthcare services. Gatekeeping—the act of determining access to specialist care and diagnostic services—is a routine task for GPs. A core goal of the gatekeeping model is to make healthcare accessible while ensuring that service delivery is feasible. Concerns about the effectiveness of GP gatekeeping

## Strengths and limitations of this study

► This study uses qualitative interviews and a hypothetical case study approach, by doing so shines important light on general practitioner (GP) decision-making processes and their perspectives on changes to referral pathways.
► The interview schedule was extensively piloted with various professionals prior to data collection and generated rich data.
► Data analysis had inductive and deductive elements building from previous review work.
► GPs were recruited through the local clinical research network (CRN). Therefore, we did not capture the practices and perspectives of GPs not actively involved in research through the CRN.
► This work is not epistemologically, or methodological positioned to comment on the effectiveness of the referral pathways.

are longstanding in the primary care literature.[1–3] Recent reviews have suggested that, in general, GP gatekeeping is linked with a better quality of care and lower service utilisation.[4] Yet questions persist about patient satisfaction with the model and the accuracy of gatekeepers in identifying certain conditions.[4] In the UK, some clinical commission groups have alleviated GPs of their gatekeeping responsibilities for specific clinical populations, including paediatrics and some mental health services.[5] This has been done by shifting gatekeeping duties to professionals in adjacent fields (eg, health visitors, social care and education) or introducing direct referral or self-referral models.

The assessment of developmental conditions such as autism spectrum disorders (autism) and attention deficit hyperactivity disorders (ADHD) reflects these broader tensions around the gatekeeping role in primary care. Referral pathways in the UK often require that GPs initiate referrals for children where there is a query of autism or

ADHD to Child and Adolescent Mental Health Services (CAMHS). Much of the research on autism and ADHD in general practice focuses on GP knowledge and attitudes towards the respective conditions.[6–9] Survey work indicates that in general GPs have a sound understanding of autism but little confidence responding to the condition.[7] Still, review work on GPs' knowledge of autism and ADHD has identified some outmoded aetiological theories still receiving endorsement.[6 10] Consequently, calls for training from GPs and researchers alike are recurrent in much of this work.

Remarkably few studies, however, have explored how GPs make these decisions in practice. This is within a context where parents often describe the pathway to diagnosis as challenging,[11–13] and reasons for delays in referral are often felt by parents to be unclear. Some insight, however, can be gleaned from a Canadian study by Kennedy et al[14] on medical students at the University of Toronto, which explores knowledge–practice discrepancies following educational programmes. In this qualitative study, the authors identified various factors, including patient motivations, systemic issues, social and clinician factors as explanations for referral decisions. Increased uncertainty and urgency, somewhat predictably, prompted referrals.[14] Indeed, clinical judgement appears to be an essential factor even within contexts where best practice guidance recommends standardised screening for developmental conditions.[15] In the UK, best practice guidance[16 17] suggests that standardised tools are not essential to identify possible autism in children, and universal screening for ADHD in is explicitly discouraged. Instead, the National Institute for Health and Care Excellence (NICE) recommends that referrers, including GPs, explore possible behavioural markers, predisposing factors (eg, family history) and obtain an account of these features across different contexts.

Our study sought to provide an account of the assessment practices some UK-based GPs engage in when identifying autism and ADHD. This research was conducted in the east of England, where changes to the configuration of local pathways mean that referrals from GPs are rarely accepted. Therefore, a subsidiary aim was to explore how GPs experience these changes and how clinicians in specialist services think about the role of GPs. Although this study takes place in a particular setting, the themes identified here might have relevance to broader national conversations about the organisation of referral pathways and the gatekeeping role of GPs.

## METHOD

The data presented in this study were collected as part of a project exploring assessment practices in healthcare professionals (n=25). Specifically, we conducted semi-structured interviews with eight GPs and seven healthcare professionals working across a neurodevelopmental team and CAMHS in an English city. Additionally, we interviewed a further 10 professionals who were working at various social and neurodevelopmental services, including national services, across the UK. Here, we report on the part of the study concerned with GPs' experiences of identifying autism and ADHD, changes to local referral pathways, and the views of specialists regarding the role of GPs in the neurodevelopmental assessment. This project was approved by the University of Cambridge Psychology Ethics Committee (PRE.2018.019), The Health Research Authority and local NHS research and development teams. All participants provided written informed consent before data collection. Consent was also provided verbally at the end of each interview.

### Referral pathway

The study was conducted in a socioeconomically diverse area in the East of England, in urban and rural areas serving a population of nearly a million people. Here, community and paediatric teams often work together to provide services for children under 5 years with a suspected developmental condition including autism and ADHD. Recent changes to the referral pathway mean that referral pathway is configured such that referrals mostly come from preschools and or health visitors, rather than GPs. For school-aged children, referrals tend to go through schools unless the child has an established neurodevelopmental condition. In the first instance, most parents are offered support in form of psychoeducation and parenting groups by neurodevelopmental team. Should questions remain about the child's development, then an assessment is conducted by the CAMHS-neurodevelopmental team. The CAMHS neurodevelopmental team is comprised of various professionals including psychologists, psychotherapists, psychiatrists, nurses, occupational therapists, speech and language therapists and paediatricians. CAMHS—community team, on the other hand, work with children with mental health problems and accept a referral from an array of sources including GPs, allied healthcare professionals, social workers and education professionals. There are also teams specialising in child safeguarding.

### Data collection

The local clinical research network (CRN) invited GPs to take part. Professionals from CAMHS, social and neurodevelopmental services were recruited using a combination of purposive, convenience and snowball sampling techniques. BC conducted all interviews either in person or remotely (eg, via telephone). Face-to-face interviews were conducted in GP practices or clinic rooms. Data were collected between January and May 2019. For further information, see table 1. Before data collection, we developed a topic guide based on existing literature and experience of the authors. The guide was piloted with three healthcare professionals working in mental health or developmental services. Questions were also discussed with two academic GPs. The final version of the guide was divided into the following sections: professional background, routine clinical work, a hypothetical

**Table 1** Participant and interview characteristics

| Participant ID | Gender | Experience (years) | Setting | Interview length (min) |
|---|---|---|---|---|
| PTGP01 | Female | >20 | Local GP practice | 43 |
| PTGP02 | Male | 4 | Local GP practice | 41 |
| PTGP03 | Female | >20 | Local GP practice | 44 |
| PTGP04 | Male | >20 | Local GP practice | 64 |
| PTGP05 | Male | 19 | Local GP practice | 29 |
| PTGP06 | Male | >20 | Local GP practice | 37 |
| PTGP07 | Male | >20 | Local GP practice | 71 |
| PTGP08 | Male | 14 | Local GP practice | 61 |
| PTND01 | Male | 17 | Local ND service | 66 |
| PTND02 | Female | >20 | Local ND service | 64 |
| PTND03 | Female | 13 | Local ND service | 58 |
| PTND04 | Female | >20 | Local ND service | 64 |
| PTND05 | Female | 14 | Local CAMHS | 69 |
| PTND06 | Female | 13 | Lifespan Autism Service | 65 |
| PTND07 | Male | 3 | Child autism service | 55 |
| PTND08 | Female | 10 | Tier 4 CAMHS | 62 |
| PTND09 | Female | 19 | Tier 4 CAMHS | 53 |
| PTND10 | Female | 10 | Tier 4 CAMHS | 58 |
| PTND11 | Female | 16 | Tier 4 CAMHS | 48 |
| PTND12 | Male | 6 | Tier 4 CAMHS | 54 |
| PTND13 | Female | >20 | Tier 4 CAMHS | 43 |
| PTND14 | Male | >20 | Tier 4 CAMHS | 55 |
| PTND15 | Female | 4 | Tier 4 CAMHS | 61 |
| PTND16 | Male | 4 | Local CAMHS | 63 |
| PTND17 | Female | >20 | Local ND | 65 |

.CAMHS, Child and Adolescent Mental Health Services; GP, general practitioner; ND, neurodevelopmental.

case study and referral pathways. See online supplemental file S1 for the hypothetical case study. The hypothetical case study and the discussions of routine clinical work were used in an effort to elicit in-depth information about clinical reasoning and assessment practices. At the beginning of each interview, participants were asked not to disclose any personally identifiable information about any patients. Questions in the section on routine clinical work were also prefaced with this reminder (see online supplemental file S2 for interview guide).

BC has experience working in a neurodevelopmental service as an assistant psychologist, where he became interested in the interaction between cognate health services. MW is a consultant clinic psychologist and RD is a social scientist. Both RD and MW are interested in assessment practices for social and neurodevelopmental conditions. AM is a placement student with an interest in child development.

We adopted an 'information power' approach to guide recruitment and sample size.[18] This approach spotlights the following considerations for establishing a sample size in qualitative research: study aim; sample specificity; established theory; quality of dialogue; and analysis strategy.[18]

### Patient and public involvement

A general patient and public review panel at a local hospital provided feedback and suggestions on the research materials, including the topic guide. This panel did not necessarily have specific experience or personal contact with Autism Spectrum Disorder or Attention Deficit Hyperactivity Disorder.

### Data analysis

Data were analysed and interpreted using the framework method outlined by Gale et al.[19] This method has the advantage of inductive and deductive elements. This allows for ideas from the existing literature to be brought together with data derived from the interviews to develop an analytical framework. This included a recent systematic review on autism in general practice[10] and a review by Tatlow-Golden et al[6] on GPs and ADHD. All interviews were transcribed by BC or a professional transcription

**Table 2** Summary and description of the main themes

| | Themes | Description |
|---|---|---|
| Identification | Explicit information | This theme describes forms of information which are considered explicit. This includes reference materials, behavioural markers and parental report. |
| | Implicit information | This theme captures forms of information which are less ostensive than material described above but nevertheless contribute to clinical decisions. This includes clinical intuition and prior knowledge of families. |
| Referral pathways | Perceptions of the new referral pathway | This theme provides an account of GPs and specialists impressions of the new pathway. |
| | Specialist views on the role of GPs | This theme describes specialists' views on the role of GPs. |
| | Information sharing as a barrier and opportunity | This theme describes participant's views on information sharing between services. |

GP, general practitioner.

service. Transcripts were read three times, and all audio recordings were listened to at least once before the first round of coding. In the initial stages, transcripts were coded using line-by-line coding. All transcripts were coded by BC, and several of the transcripts were also read in full by AM and RD. Regular meetings were held between the authors to discuss the data. All authors approved the final series of themes. Transcripts were coded by hand, and data were organised and grouped using flashcards. Here, results pertaining to identification and referral pathways are discussed. Specialist neurodevelopmental assessment practices and differential conceptualisation are explored elsewhere. Prior to submission, participants were each sent the results and offered the opportunity to comment on the findings.

## RESULTS
The findings are presented in two sections. The first section focuses on the methods and sources of information the GPs (n=8) used when screening possible autism and ADHD in children. The second section discusses material from the entire set of transcripts (n=25) to explore a range of perspectives on changes to the pathway and the role of the GP. A summary of the main themes is presented in table 2.

## Identification
There was some variation regarding the methods and techniques used by GPs to identify autism and ADHD in children. References to a diverse array of different forms of information could be seen across the transcripts, including both tacit and explicit sources. These include various clinical or behavioural markers, unstructured behavioural tasks (eg, 'pointing to assess joint attention' task, prior knowledge of the family and discussions with colleagues) and personal experience. Nevertheless, the extent to which GPs considered, used and triangulated this information varied considerably, with some GPs offering to contact schools and others basing the referral on parental report.

Explicit information: an assortment of diagnostic or clinical markers for each condition were described by participants. Oft cited features of autism included atypical eye-contact, delayed language, fixed or specialised interests (eg, Emergency Departments), ritualistic behaviours (eg, rocking) and sensory sensitives. When thinking about ADHD, most practitioners characterised the condition by inattention, problems with concentration, impulsivity, social problems and impaired academic functioning. Yet some GPs expressed uncertainty and hesitancy when asked about particular indicators:

Early markers? I'd probably have to look it all up, actually……And often I do. When I've got a patient coming in, I just have a sort of screen what the most common symptoms PTGP02

There's gonna[sic] be diagnostic criteria for that but don't ask me what they are. There's a big long list of diagnostic criteria, but I kind of think that's more a specialist job to apply the diagnostic criteria in detail before making the diagnosis, but I'd probably spot the warning signs as it were and refer on as appropriate. PTGP06

And indeed, several practitioners described looking up markers using professional sources such as GP Notebook, Clinical Knowledge Summaries or Patient.co.uk as well as some lay sources including Google or Wikipedia to find specific behavioural markers. Of note, GPs did not refer to NICE guidance.

In general, however, GPs appeared to agree on the importance of parental report. This is, of course, understandable as parental concerns are an essential component of the formal assessment for many behaviourally diagnosed developmental conditions. While describing past cases, one GP commented:

Nine-tenths is the story you're given by the parents. Because they are the…… as I say to parents, you know your son or daughter better than anybody in the world. So, we have to listen to what they have to say, [and their] ideas, concerns, and expectations PTGP07

And indeed, the majority of participants expressed similar sentiments. Importantly, however, most GPs indicated that parental report alone was not sufficient grounds for a referral. Instead, it was suggested that such reports should be corroborated with observations of the

child. Yet when facing uncertainty, approaches varied. For instance, after reflecting on complex or uncertain cases, one GP remarked:

> Just got to go with what the parents are thinking PTGP06

However, another GP was especially concerned with diagnostic trends and the medicalisation of non-medical behaviours. For this GP, it was particularly important to triangulate parental concerns, observations of the child and reports from the child's school. This GP reflected on a case where parents queried a diagnosis of ADHD following conversations with a family friend:

> Speaking to the friend caused them to say maybe he [the child] has got ADHD. But in actual fact, I really don't think he has, and the last thing you'd want is for this kid to go on unnecessary medication PTGP04

He went on to explain that after receiving consent from the child's parents to contact the child's school:

> [I] spoke to his teacher and actually this was an example of where the school actually had a really good handle on him. The teacher said he's a lovely kid… He just isn't set up for rules so there was nothing he's doing at school that would make me worried. He's a lovely lad, and you can engage him, and he can concentrate and focus when he wants to PTGP04

In contrast, however, there was a least one instance where a GP's decision to refer seemed to be based predominantly on parental insistence, rather than clinical observations or judgement:

> [Refers to another family member] seemed to know it all. [Parent] was saying that they thought the child had autism on the basis that [the child is] behind with learning, not reading and writing yet, didn't like social situations…[…]…. And they said that the school didn't think the child had autism. So, I have referred… I mean [the child] seemed normal, sat doing not a lot, but seemed normal. PTGP05

Subsequently, this participant indicated that the chances of the referral being rejected were '100%' due to the configuration of local referral pathways. When this happens, he explained he would urge the parents to go back to the school.

Tacit information: GPs also often drew implicitly from the language of folk psychology regarding typical and atypical child development. Phrases such as 'a little odd', 'just isn't what most children do' or 'clashes with normal expectations' can be found throughout the data. These were often used in reference to a specific marker or behaviour, such as 'rituals and behaviours that weren't quite in keeping with a normal child of her age'. Here, the term tacit knowledge is used broadly to refer to practical or soft knowledge that is not easily quantifiable.

Clinical intuition was important for deciding between typical and atypical development, but at times, challenging to articulate:

> As a GP you get a subconscious idea of the spectrum of the range with children - from the kid who'll sit there like butter wouldn't melt in their mouth, like a bit oddly so, to the kid who's climbing up your curtains. [And] You get a feel of parental interaction, with 'you stop doing that now I've told you before' to the parent who just watches the child smash your ophthalmoscope' PTGP04

> I think it's difficult, sometimes, to describe what turns into a kind of sixth sense. Really you get a clue, don't you? And sort of that kind of gut feeling, but it is about the behaviour. PTGP07

Prior knowledge or experience with specific children and families was also crucial for several GPs. When reflecting on cases, it was not uncommon for practitioners to preface conceptualisations with remarks such as 'I've known him since… well antenatally', 'I know the family' or '[Mum/Dad] is also my patient'. This seemed to offer a degree of context and explanation for the child's presentation. For instance, when describing children with a query of a neurodevelopmental condition, some GPs remarked on traits they had seen in other family members or diagnoses of other family members they were aware of.

GPs were also attuned to socioenvironmental or parenting factors that might be contributing to the child's presenting symptoms such as discrete participating events, parental separation or conflict. Having this overview of the patient was, for many, one of the core strengths of general practice:

> I suppose this is where Family Medicine really comes into its fore, isn't it? Because they're [both child and parents] usually, not always, but usually all our patients. So, sometimes we have this interesting dilemma about whom is the patient. PTGP07

Yet this expertise, some felt, was not always appreciated by colleagues in specialist services. When reflecting on the experience of having referrals rejected, one GP remarked:

> I sometimes wonder whether they [specialist assessment service] actually consider the family factors that we know of that we write in out letters PTGP03

### Perceptions of the new referral pathway

Most of the GPs were aware of the changes to the assessment pathway that meant referrals for neurodevelopmental assessment typically come through schools or health visitors. However, it was unclear whether two GPs were aware of these changes. Three stances to these changes were identified in the data: accepting, ambivalent and critical. Practitioners who were more accepting of the changes tended to reason that schools are better positioned to identify such developmental conditions:

Well, you see I think community paediatrics probably has a point. Because small child gets brought in to see the doctor and they're looking around looking reasonably normal but what do I know. Whereas the school and other people that interact with the child over a long period of time are in a better position to make an assessment than me. PTGP05

They'll [schools] be better at recognising it than me, so I'm happy, doesn't matter where the referral comes from, as long as it happens in a timely way it doesn't have to come from a GP. PTGP06

One participant had a more ambivalent attitude. For this participant, there was an acknowledgement that schools are often well-placed to identify atypical development. Still, they maintained that limiting the ability of GPs to refer put them in a challenging position:

Difficult. Because you can see the logic in that, actually there's so much more to this than having a name put to your child's odd behaviour very few of these children will benefit from something medical…[…]… the problem comes really when a parent comes in and says I've been to the teacher three times, and the teacher says they think he's fine and if you're really worried you can go and see your GP. Because you've no idea did the teacher really say that. PTGP04

While one GP was critical of the pathway:

Just a disaster, just a road crash really - trying to get children seen with developmental or behavioural problems is increasingly difficult, and in fact, for many patients, we end up having to go if they're school age we end up having to go through school…[…]… And that's a real nightmare for me because it means I'm having to delegate that to a third party who is not actually a health service PTGP07

This GP felt that the pathway was also a threat to professional status and identity, reflecting a devaluation of primary care.

Professionals in the neurodevelopmental services tended to view the changes as positive. Professionals in the neurodevelopmental team reported on the impact these changes have had on service-level pressures, including waiting times for assessment:

We've got the shortest waiting times for assessment for autism and ADHD. Less than eighteen weeks, whereas they were eighteen months to two to three years [before] PTND01

### Specialist views on the role of GPs

When asked directly whether GPs had a role in identifying developmental conditions, most specialists indicated that there was indeed a role for GPs. Yet this was often couched with an array of caveats about professional and organisational barriers to identification. The most common

barrier, according to the specialists, was the duration of primary care consultations and a lack of training or knowledge about neurodevelopmental conditions:

They need to be given more time to do it properly and more training. They get very little training at all really but if they got proper training and given a bit more time. Even fifteen-twenty minutes, but at the moment all they could do is to at least know the NICE guidance and know what are the signs and symptoms and take a detailed history and follow the local pathway really. Clearly, if we have GPs with a special interest in children, they got better training, and clearly, they have a lot of role to play with the ADHD medication shared care and those kinds of things. PTND01

At best, what they should do is make good referrals to specialist teams. But beyond that, I don't know if it would be useful for people who are under massive strain and pressure and who have like whatever is it eight to ten-minute appointments, I hear that's the average, but I've never had any more than six min really, so I mean I don't know how you could do anything bar account for the family's request and signpost them to the appropriate teams. PTND07

A lack of training was also framed as problematic by one GP:

I think also in terms of what we get taught. It may change now obviously. I trained thirty years ago literally we had no training at all…[…]… We'd all heard of autism but everything I know about neurodevelopmental disorders, not that there's much of it, has been acquired post-grad. PTGP04

References to the time afforded for consultations can also be found throughout the GP interviews. For some, this was felt to be a significant barrier to identification. To circumnavigate some of these challenges, one GP described bringing families back for multiple consultations.

Information sharing as a barrier and opportunity:

Another topic that runs through the data is the importance of informal networks and the issues with sharing information between services. In general, informal networks could be described as internal and external. Internal networks mostly consisted of practice staff, including administrative staff, GP colleagues and nurses. By contrast, external networks consisted of educational professionals and colleagues in secondary care. Due to the reconfiguration of primary care services, health visitors seemed to occupy a position between these two networks:

We used to have Health Visitors attached to the practice, but they don't exist anymore. I don't know who our Health Visitor is. I've never met them. PTGP05

Meanwhile, most GPs acknowledged that nursery staff, primary school teachers and other educational

professionals were essential sources of information when thinking about child development. Yet the lack of a linked system for educational and primary care records presented challenges in terms of sharing this information:

> We have occasional contact with schools but not very much. Not often. I'd be unsure about the boundaries and confidentiality and things like that, to be honest. PTGP05

There also seemed to be a lack of communication between GPs and specialist services:

> It's so difficult because you know you'll write the letter, but you don't know if they'll actually get any help or whether they'll get put on the waiting list or whether someone else will monitor the child. So that's the tricky bit, really. PTGP03

> let's say we're querying autism they [neurodevelopmental team] would send the referral back. And say it needs to be referred through the school which is quite doable because quite often they have started with the school. And the school have said have you seen your GP and of course then it looks like passing the parcel. PTGP04

## DISCUSSION
### Summary
GPs used tacit and explicit forms of information when identifying autism and ADHD in children. These included clinical or behavioural markers, parental report, prior knowledge of the child and family and professional networks. For most, parental concerns were the chief factor driving referral decisions. However, a few participants described instances where they had sought information from other sources (eg, schools). Nevertheless, changes to the configuration of local pathways have meant that referrals from GPs for neurodevelopmental assessment are now rarely accepted. GPs had mixed views on these changes. Most specialists agreed that GPs did have a role in identifying neurodevelopmental conditions yet expressed concerns about a perceived lack of training or knowledge and framed time pressures as problematic.

### Strengths and limitations
The current study adds to our understanding of early identification by gleaning the perspectives of GPs and those in specialists' assessment services. From a methodological perspective, the flexible interview guide and the combination of case-based discussions and hypothetical case study allowed us to elicit rich narratives about these topics. Further, by analysing discourses of past and hypothetical cases, we were able to explore some of the other forms of knowledge that come into play. Additionally, our study was conducted in a setting where GPs have been, to a large extent, absolved of their gatekeeping responsibilities for identifying autism and ADHD in children. Therefore, the current study presents a unique opportunity to explore how GPs experience having a reduced role for a specific patient group and thus adds to national conversations about the nature and future of general practice. That said, it is essential to consider whether the findings about identification are transferable to other contexts. Regarding identification, given that specialists espoused similar issues with referrals in different settings, it seems unlikely that the methods and techniques used by GPs in this area were atypical. As recruitment of GPs was completed through the local CRN, it is not possible to determine how many GPs decided not to take part in the study. This might raise other concerns about the representativeness of the GP sample. However, as the analysis illustrates, there was considerable diversity in the views and opinions expressed by the GPs. Another limitation of this study is that, although data were discussed at regular meetings between the research group, BC conducted and coded the analysis. As per the method, BC has previously worked in a neurodevelopmental service. To address possible issues with research bias, BC wrote reflections throughout the process and engaged in peer and academic supervision. Finally, this research took place prior to COVID-19 pandemic. Therefore, with GPs under considerable strain, it is important to consider whether how the pandemic might have shaped referral pathways and indeed GP's attitudes towards identifying neurodevelopmental conditions.

### Comparison with the literature
Most studies about GP knowledge of autism and ADHD have focused on explicit knowledge of clinical markers.[6 7 9 20] Yet, as others have shown, clinical judgement is core to referral decisions.[14 21] Naturally, knowledge of clinical markers is important for identifying these conditions. For ADHD, some codeveloped training tools are showing promise.[22] Still, an overemphasis on this form of knowledge risks driving attention away from the other sources GPs draw on, including prior experience with the child or family. Our study, therefore, adds to the understanding of identification by tracing out the various forms of explicit and tacit material which GPs draw on when determining whether a child requires formal assessment.

Several studies have identified that GPs frequently report having little training in autism,[7] and ADHD.[6] It follows that more training could be helpful. Our data lend some support to these findings, and broadly speaking, we agree with these calls for more training. The 'lack of training' thread runs throughout the primary care literature. However, a degree of caution is warranted, as framing the problem as one of 'a lack of training' risks (1) flattening the conceptual complexity associated with identifying these conditions, (2) silencing the host of organisational shortcomings that make referral decisions challenging and (3) camouflaging alternate solutions such as the integration of health, educational, justice or social care records or changing pathways.

## Implications for research and practice

Elsewhere, questions have been raised about GP gate-keeping.[1–3] As such, zooming in on a particular pathway means that we were able to explore in detail how those on the ground experienced changes to GP gatekeeping. It might be envisaged that GPs would welcome changes that reduce some of the pressure on them. Yet, GPs in this study expressed mixed views. In contrast, specialists tended to view the changes positively and credit these changes with preventing saturation of the service. Our research is not positioned to explore the impact that these changes have on service delivery. We recommend that future work explores how such changes impact patient satisfaction, waiting lists and numbers of accepted referrals. It will be also important to consider the unmet needs of children who do not receive access to services.

Issues around the quality of GP referrals ran through the specialist interviews. As such, we anticipate that the analysis of autism and ADHD referrals using health records might yield further insights into the level and quality of information required by specialist services.

Finally, it seems likely that GPs in most settings will retain gatekeeping responsibilities for autism and ADHD for the foreseeable future. The findings indicate that some GPs used lay sources such as Google or Wikipedia. As such, we recommended that future work further explores the modes of professional and lay information used by GPs to inform their clinical decision-making. In particular, we would welcome research that explores whether the forms of information used by GPs have an impact on referral decisions and on referral acceptance.

**Contributors** BC, MW and RD contributed to the conceptualisation and design of the study. BC applied for governance and ethical approvals, and collected the data. BC conducted the initial coding while RD, AM and MW contributed to analysis of the data. Each author offered interpretations of the findings. The final set of themes were agreed by each of the authors. BC wrote the first draft of the manuscript. MW, AM and MW provided critical feedback and suggestions on subsequent drafts. All authors contributed to and approved the final manuscript. BC is acting as guarantor for this work.

**Funding** The authors wish to thank NIHR School for Primary Care Research [RG94577] for their support for work on this paper. This research was also funded in whole, or in part, by the Wellcome Trust [WT103343MA]. For the purpose of open access, the author has applied a CC BY public copyright licence to any Author Accepted Manuscript version arising from this submission. We would also like to the CRN for help with recruitment. Warm thanks are extended to authors Prof Marinus van IJzendoorn for feedback on a draft of the manuscript.

**Disclaimer** The views expressed are those of the authors and not necessarily those of the CRN, Wellcome, NHS, the NIHR or the Department of Health.

**Competing interests** None declared.

**Patient and public involvement** Patients and/or the public were involved in the design, or conduct, or reporting, or dissemination plans of this research. Refer to the Methods section for further details.

**Patient consent for publication** Not applicable.

**Ethics approval** This project was approved by the University of Cambridge Psychology Ethics Committee [PRE.2018.019]. Participants gave informed consent to participate in the study before taking part.

**Provenance and peer review** Not commissioned; externally peer reviewed.

**Data availability statement** No data are available. Although all participants were reminded not to disclose any personally identifiable information about patients or families, the transcripts do include reflections on routine clinical work and service arrangements. Thus, to further safeguard the privacy or the participants and those involved in their services, we cannot make the transcripts available. Please contact the authors for further details on the data.

**ORCID iDs**
Barry Coughlan http://orcid.org/0000-0002-1484-6491
Alissa Mann http://orcid.org/0000-0002-3618-0395
Robbie Duschinsky http://orcid.org/0000-0003-2023-5328

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
