## [Reviewer comments · BMJ Open]

ARTICLE DETAILS

TITLE (PROVISIONAL)	Clinical perspectives on the identification of neurodevelopmental conditions in children and changes in referral pathways: Qualitative interviews.
AUTHORS	Coughlan, Barry; Woolgar, Matt; Mann, Alissa; Duschinsky, Robbie

VERSION 1 – REVIEW

REVIEWER	Charlotte Hall Institute of Mental Health, NIHR CLAHRC-East Midlands, Psychiatry
REVIEW RETURNED	10-Mar-2021

GENERAL COMMENTS	This study qualitatively explored how GPs identify ADHD and ASD in their clinical practice in a CRN footprint within the UK. GPs reported using different sources of information to inform their decision to refer patients for specialist services, including gaining information from the family and consulting expert and lay resources. The findings support previous studies that show GPs require more training in identifying ADHD. Overall, the manuscript is well written and provides an important contribution to the current, growing literature within this field. I have the following specific comments; - Spell out GPs for a non-UK audience- A table summarising the main themes would be useful for the results section – one summarising methods and sources of information used, and one based on the rest of the analysis perhaps?- In the limitations it would be worth reflecting that BC has experience of working in NDD and conducted and coded the interviews – reflect on researcher bias- It would be worth reflecting in your discussion how the views found from GPs reflect families views on the ADHD/ASD care pathway and their experiences of getting a referral to specialist services- On page 16 (discussion) you mention that GPs perceived lack of training/knowledge and length of time as problematic – it isn't clear to me if that is referring to length of time to be involved in training, or length of time from referral to diagnosis as problematic- Although perhaps beyond the scope of this study it would be interesting to reflect on whether GPs that draw on specific information are more/less likely to get their referrals accepted than others – knowing this may also be informative for other GPs- There have recently been papers exploring improving GPs ability to identify ADHD, it may be worth reflecting on those: French, B., et al (2020). Development and evaluation of an online education tool on attention deficit hyperactivity disorder for general practitioners: the important contribution of co-production. BMC Family Practice, 21(1), 1-10.
--

	French, B., et al (2020). Assessing the efficacy of online ADHD awareness training in primary care: pilot randomised control trial evaluation with nested qualitative interviews. JMIR Med Educ. - The conclusion, line 47 I think the sentence starting “the findings that some GPs” is incomplete
--	--

REVIEWER	Peter Rosenbaum McMaster University, Paediatrics
REVIEW RETURNED	24-May-2021

GENERAL COMMENTS	This was a challenging paper to review. First, the specific research question is somewhat unclear. After reading the full text, I am not convinced that this is work with generalizable findings. To me the report reads more as a quality assurance or quality improvement exercise, designed to evaluate changes in the referral system in the community in which the work was undertaken. That does not invalidate it, but does raise the issue of whether this should be a journal article or a report to the several actual and potential players in the arena of childhood disability and impairment. It will be apparent that GPs exist within a bigger system of protagonists - families, school systems, specialists, and so on. What the GSs do, how they do it, and how well they do it (not discussed in this report) all depend on issues such as time (discussed briefly) and the amount of support and connectedness they have with the other people in the community who might contribute to the issues with which GPs are expected to deal. If specialists are critical of GPs' decision-making, then GPs are likely to behave and 'perform' differently than if they feel supported. Were there better-integrated interlinkages between schools, families, GPs and specialists, the whole system would presumably work more effectively. This is lacking in many communities, and is not meant as a critique of the community in which this work was done. In summary, the findings of this study should be shared with the many interested parties in the community in which the work was done. I am not convinced that there are broadly applicable findings for the world at large. Finally, I wish to note that there are a number of distracting typos or other writing issues that disrupted the flow of the text - and hopefully these have been captured and highlighted for revision. The reviewer provided a marked copy with additional comments. Please contact the publisher for full details.
--

VERSION 1 – AUTHOR RESPONSE

Reviewer: 1

Dr. Charlotte Hall, Institute of Mental Health, NIHR CLAHRC-East Midlands

Comments to the Author:

This study qualitatively explored how GPs identify ADHD and ASD in their clinical practice in a CRN footprint within the UK. GPs reported using different sources of information to inform their decision to refer patients for specialist services, including gaining information from the family and consulting expert and lay resources. The findings support previous studies that show GPs require more training in identifying ADHD.

Overall, the manuscript is well written and provides an important contribution to the current, growing literature within this field.

I have the following specific comments;

- Spell out GPs for a non-UK audience

Thank you for this helpful comment. We have now included an introductory sentence explaining GPs for non UK audiences.

For reference the new sentence reads:

In the UK General Practitioners (hereafter GPs) are one of the main providers of primary healthcare services

- A table summarising the main themes would be useful for the results section – one summarising methods and sources of information used, and one based on the rest of the analysis perhaps?

Thank you for this helpful suggestion. We completely agree and thus have added another table summarising and describing the main themes. Since interviews were the only form of data used we would gently ask we continue to explain the methods and information in a textual format.

- In the limitations it would be worth reflecting that BC has experience of working in NDD and conducted and coded the interviews – reflect on researcher bias

Thank you for this helpful comment. I have added discussion of this in the limitations section.

For reference, the text reads:

Finally, another limitation of this study is that, although data were discussed at regular meetings between the research group, BC conducted and coded the analysis. As per the method, BC has previously worked in a neurodevelopmental service. To address possible issues with research bias, BC wrote reflections throughout the process and engaged in peer and academic supervision.

- It would be worth reflecting in your discussion how the views found from GPs reflect families views on the ADHD/ASD care pathway and their experiences of getting a referral to specialist services

We completely agree that this is an interesting question. Unfortunately, the restrictions on the word count means we are limited in the amount of discussion we can add. Considering we don't have data on parents views we feel that the other material is more closely linked to the data.

- On page 16 (discussion) you mention that GPs perceived lack of training/knowledge and length of time as problematic – it isn't clear to me if that is referring to length of time to be involved in training, or length of time from referral to diagnosis as problematic

Apologies that was a typo it was meant to be lack, not length. We have made a minor adjustment to make the sentence clearer. For reference:

Most specialists agreed that GPs did have a role in identifying neurodevelopmental conditions yet expressed concerns about a perceived lack of training or knowledge and framed time pressures as problematic.

- Although perhaps beyond the scope of this study it would be interesting to reflect on whether GPs that draw on specific information are more/less likely to get their referrals accepted than others – knowing this may also be informative for other GPs

We think this is such an interesting question, but unfortunately we are not methodologically position to offer anything more than pure speculation. However, we think this is an urgent question and as such have added the following line to the conclusion:

“we would welcome research that explores whether the forms of information used by GPs has an impact on referral decisions and on referral acceptance.”

- There have recently been papers exploring improving GPs ability to identify ADHD, it may be worth reflecting on those:

French, B., et al (2020). Development and evaluation of an online education tool on attention deficit hyperactivity disorder for general practitioners: the important contribution of co-production. *BMC Family Practice*, 21(1), 1-10.

French, B., et al (2020). Assessing the efficacy of online ADHD awareness training in primary care: pilot randomised control trial evaluation with nested qualitative interviews. *JMIR Med Educ*.

- The conclusion, line 47 I think the sentence starting “the findings that some GPs” is incomplete

Thank you so much for these helpful references. We have included a sentence directing the reader to this work:

For ADHD, some co-developed training tools are showing promise 20

Reviewer: 2

Prof. Peter Rosenbaum, McMaster University

Comments to the Author:

This was a challenging paper to review. First, the specific research question is somewhat unclear.

Thank you for taking the time to read and feedback on our paper. We agree that the research question could be clearer. As such we have amended the text to

“Our study sought to provide an account of the assessment practices some UK-based GPs engage in when identifying autism and ADHD. This research takes place in an English city where changes to the configuration of local pathways mean that referrals from GPs are rarely accepted. Therefore, a subsidiary aim was to explore how GPs experience these changes and also how clinicians in specialist services think about the role of GPs”

After reading the full text, I am not convinced that this is work with generalizable findings. To me the report reads more as a quality assurance or quality improvement exercise, designed to evaluate changes in the referral system in the community in which the work was undertaken. That does not invalidate it, but does raise the issue of whether this should be a journal article or a report to the several actual and potential players in the arena of childhood disability and impairment.

Thank you for these reflections. Given the nature of the study (qualitative) it is difficult to generalise the findings to other settings and we discuss particular challenges in the limitations. However, we hope this research contains some important insights and findings about ASD and ADHD assessment practices in general practice and also on the configuration of primary care services. Naturally there is wide variation in primary care systems throughout the world, but still we think this research contains important insights about identifying ASD/ADHD and service pathways.

It will be apparent that GPs exist within a bigger system of protagonists - families, school systems, specialists, and so on. What the GPs do, how they do it, and how well they do it (not discussed in this report) all depend on issues such as time (discussed briefly) and the amount of support and connectedness they have with the other people in the community who might contribute to the issues with which GPs are expected to deal. If specialists are critical of GPs' decision-making, then GPs are likely to behave and 'perform' differently than if they feel supported. Were there better-integrated interlinkages between schools, families, GPs and specialists, the whole system would presumably work more effectively. This is lacking in many communities, and is not meant as a critique of the community in which this work was done.

Thank you for this comment. We completely agree with this point and information sharing between services is a main theme. In the discussion we identify some potential issues with the lack of training refrain however we do not have the word count to include further discussion.

In summary, the findings of this study should be shared with the many interested parties in the community in which the work was done. I am not convinced that there are broadly applicable findings for the world at large.

Thank you for this comment. We acknowledge that findings might not be transferable to the world at large. However, we do not think that the primary aim of qualitative research is to generalise to the world at large. Rather we think the aim of qualitative research is to make sense of certain phenomena or experience. We think the issues around the GPs assessment practices, their role, and changes to gatekeeping systems will be of a primary care audience and the readers of BMJ Open.

Finally, I wish to note that there are a number of distracting typos or other writing issues that disrupted the flow of the text - and hopefully these have been captured and highlighted for revision.

Thank you for flagging these issues. We have tried to address these and all changes are marked in track changes.

VERSION 2 – REVIEW

REVIEWER	Charlotte Hall Institute of Mental Health, NIHR CLAHRC-East Midlands, Psychiatry
REVIEW RETURNED	30-Sep-2021

GENERAL COMMENTS	This study aimed to explore 1) how GPs identify ADHD/ASD and how they view changes in a local care pathway. The paper is generally well written, but there are some tense changes throughout which need careful examination. The method is at times confusing, with mentions of case studies/vignettes which aren't accounted for fully in the methods (or the results). It isn't always clear what the initial pathway was and what the change now is (a diagram may help with this) and how generalisable this is to the rest of the UK. Some services have also undergone change in pathways as a result of COVID, so a reflection on how relevant these findings are would be important. The authors could more explore existing literature that is in this field (some refs provided, but more are available). The study is limited to inclusion only to one City. Referral pathways vary across England and the generalisability of these findings are limited by inclusion of only one place. Given that this is an area which has already undergone some previous research, the additional value of this paper is small, existing research has demonstrated some of the findings. Abstract: Line 18 - Under objective, I had to infer that by "accepted" you meant accepted by CAMHS/paeds (?). Please make this more clear. - Also on line 18, there is a change in style, using "we" when the rest in in third person Line 26: by specialists in local services – are you referring to clinical specialists in CAMHS/paeds? - I realise space is of the essence but is it possible to summarise the
--

	changes in the pathway? Are you referring to things such as SPA? Or re-direction of identification to other professionals? Limitations (p2)  - GP missing “s” GPs - I am not clear what non-CRN GPs means? Wouldn't all GP services fall under a CRN? Perhaps this is my misunderstanding? Introduction  - Page 3 : I think the introduction would benefit from an understanding of where these referrals from GPs are too. IE. CAMHS/community paed's? - I think a mention to NICE guidelines for these disorders is also needed - Page 4 – line 29: you mention “standard assessment” but don't detail what this is/was - Line 41 – tense change “takes place” (not past tense) - It also isn't clear here who the referrals go to in this local city Method (P5)  - Was the city in north/south, more contextual information about this would be better, what population size did it serve, rural/urban and if known locally number of referrals for ADHD/ASD. - What was provided to parents – psychoeducation? How? And by whom - Line 36: where does Neurodevelopmental team sit? Which service and what is it made up on (staff). - I think a diagram of the pathway would be helpful - You state in the discussion that you use case studies and clinical vignette – but this isn't clear in the methods. It focuses on the interview guide and a “hypothetical” case study – which doesn't feel the same as saying you used case studies. Results  - P8 starting line 42: I don't really gain much insight into what behavioural marks or tasks were used. What professional networks were important? You said it varied regarding triangulation, but what was the variation – did some not do it all? How was this judged? - I think the results would also benefit by reflecting on NICE guidelines and how what was said links with this - Previous research has also mentioned about professionals feeling parent do not want a non-ADHD diagnosis (Study of user experience of an objective test (QbTest) to aid ADHD assessment and medication management: a multi-methods approach SpringerLink) - Likewise other studies have looked at training GPs for ADHD awareness XXXXX Discussion  - The Interviews were carried out pre-COIVD, the authors should reflect on how services may have changed as a result of this and the relevance of their findings - Other research has explored ways of improving GP knowledge of ADHD: https://mededu.jmir.org/2020/2/e19871/
--	---

VERSION 2 – AUTHOR RESPONSE

Reviewer 1

1 This study aimed to explore 1) how GPs identify ADHD/ASD and how they view changes in a local care pathway. The paper is generally well written, but there are some tense changes throughout which need careful examination.

Thank you for flagging the issue with tense changes. We have proofed the document and made changes as required.

2. The method is at times confusing, with mentions of case studies/vignettes which aren't accounted for fully in the methods (or the results).

We have now added a sentence describing the rationale for the case study and the discussions of routine clinical work.

For reference:

"The case study and the discussions of routine clinical work were used in an effort to elicit in-depth information about clinical reasoning and assessment practices." P6

The case study and full interview schedule are provided in the supplementary materials. We feel that considering these two methods in conjunction provides a more thorough account of assessment practices.

3. It isn't always clear what the initial pathway was and what the change now is (a diagram may help with this) and how generalisable this is to the rest of the UK.

Thank you for raising this important point. We have added further clarification regarding the referral pathway and that the change has been about the main source of referrals.

For reference this section now reads:

"The study was conducted in a socioeconomically diverse area in the East of England, in urban and rural areas serving a population of nearly a million people. Here, community and paediatric teams often work together to provide services for children under five years with a suspected developmental condition including autism and ADHD. Recent changes to the referral pathway mean that referral pathway is configured such that referrals mostly come from preschools and or health visitors, rather than GPs. For school-aged children, referrals tend to go through schools unless the child has an established neurodevelopmental condition. In the first instance, most parents are offered support in form of parenting support and psychoeducation. Should questions remain about the child's development, then an assessment is conducted by the CAMHS-neurodevelopmental team. The CAMHS neurodevelopmental team is comprised of various professionals including psychologists, psychotherapists, psychiatrists, nurses, occupational therapists, speech and language therapists and paediatricians. CAMHS- community team, on the other hand, work with children with mental health problems and accept a referral from an array of sources including GPs, allied healthcare professionals, social workers, and education professionals. There are also teams specialising in child safeguarding." P5

4. Some services have also undergone change in pathways as a result of COVID, so a reflection on how relevant these findings are would be important.

Thank you for raising this crucial point about Covid. In response, we have added the following two sentences to the discussion:

“Finally, this research took place prior to Covid-19 pandemic. Therefore, with GPs under consider strain, it is important to consider whether how the pandemic might have shaped referral pathways and indeed GP’s attitudes towards identifying neurodevelopmental conditions.” P16

5. The authors could more explore existing literature that is in this field (some refs provided, but more are available).

Thank you for this comment. We completely agree that more references are available. We do cite our review on ASD in general practice and Mimi-Tatlow’s review on ADHD, both of which do provide more complete accounts of the literature. However, as you have noted, space is of the essence and we do not feel there is sufficient room for further discussion of the literature without cutting the results.

6. The study is limited to inclusion only to one City. Referral pathways vary across England and the generalisability of these findings are limited by inclusion of only one place. Given that this is an area which has already undergone some previous research, the additional value of this paper is small, existing research has demonstrated some of the findings.

We acknowledge this as a limitation of the study. Still, we contend that this research makes a material contribution to the literature on assessment practices and service configurations. To our knowledge this is one of the first studies to explore GP assessment practices and their attitudes to having gatekeeping responsibilities alleviated for neurodevelopmental conditions. This is not a well-researched area and therefore we think this study makes a valuable contribution and might aid hypotheses for future quantitative work.

7. Abstract:

Line 18 - Under objective, I had to infer that by “accepted” you meant accepted by CAMHS/paeds (?). Please make this more clear.

Thank you for this comment. We have now adjusted this sentence to clarify that we mean referral to the neurodevelopmental team.

For reference:

“This study aimed to explore how GPs identify these conditions in practice and their perspectives on recent changes to local referral pathways that mean GP referrals to the neurodevelopmental team are rarely accepted.”P2

- **Also on line 18, there is a change in style, using “we” when the rest is in third person**

Thank you for this comment. We have now changed this to third person.

Line 26: by specialists in local services – are you referring to clinical specialists in CAMHS/paeds?

Thank you for this comment. We agree that this was unclear and therefore we have rephrased to CAMHS specialists for clarity throughout the abstract.

- **I realise space is of the essence but is it possible to summarise the changes in the pathway? Are you referring to things such as SPA? Or re-direction of identification to other professionals?**

We have confirmed that the pathway was changed so that referrals come through preschool and health visitors rather than through GPs.

For reference:

“Recent changes to the referral pathway mean that referral pathway is configured such that referrals mostly come from preschools and or health visitors, rather than GPs.”

8. Limitations (p2)

- GP missing “s” GPs

Thank you for flagging this. We have changed this to say “GPs”.

- I am not clear what non-CRN GPs means? Wouldn't all GP services fall under a CRN? Perhaps this is my misunderstanding?

Thank you for flagging this and of course you are right. We have reworded this to clarify our meaning:

For reference

“GPs were recruited through the local Clinical Research Network (CRN). Therefore, we did not capture the practices and perspectives of GPs who are not actively involved in research in the CRN.”

9. Introduction

- Page 3 : I think the introduction would benefit from an understanding of where these referrals from GPs are too. IE. CAMHS/community paed?

Thank you for this important comment we have confirmed at the topic of page 4.

For reference:

“Referral pathways in the UK often require that GPs initiate referrals for children where there is a query of autism or ADHD to Child and Adolescent Mental Health Services (CAMHS).”

- I think a mention to NICE guidelines for these disorders is also needed

Thank you for this comment. We have added a short paragraph on NICE guidance for autism and ADHD.

For reference:

“In the UK, best practice guidance^{16 17} suggests that standardised tools are not essential to identify possible autism in children, and universal screening for ADHD in is explicitly discouraged. Instead, the National Institute for Health and Care Excellence (NICE) recommends that referrers, including GPs, explore possible behavioural markers, predisposing factors (e.g., family history), and obtain an account of these features across different contexts.” P 4

- Page 4 – line 29: you mention “standard assessment” but don’t detail what this is/was

We have cut this discussion in order to make room for discussion of NICE guidance.

- Line 41 – tense change “takes place” (not past tense)

Changed to “was conducted”. P4

- It also isn’t clear here who the referrals go to in this local city

We have attempted to outline this on page 5.

For reference

“Should questions remain about the child’s development, then an assessment is conducted by the CAMHS-neurodevelopmental team. CAMHS- community team, on the other hand, work with children with mental health problems and accept a referral from an array of sources including GPs, allied healthcare professionals, social workers, and education professionals. There are also teams specialising in child safeguarding.”

10 Method (P5)

- **Was the city in north/south, more contextual information about this would be better, what population size did it serve, rural/urban and if known locally number of referrals for ADHD/ASD.**

Thank you for this helpful comment. We added the following contextual information:

“The study was conducted in a socioeconomically diverse area in the East of England, in urban and rural areas serving a population of nearly a million people.”P5

- **What was provided to parents – psychoeducation? How? And by whom**

Thank you for this – we have confirmed this

For reference

“In the first instance, most parents are offered support in form of psychoeducation in form of psychoeducation and parenting groups by neurodevelopmental team” P5

- **Line 36: where does Neurodevelopmental team sit? Which service and what is it made up on (staff).**

Thank you for flagging this. We have now clarified that the team sits in CAMHS and is made up of various professionals.

For reference:

“The CAMHS neurodevelopmental team is comprised of various professionals including psychologists, psychotherapists, psychiatrists, nurses, occupational therapists, speech and language therapists and paediatricians.”

- **I think a diagram of the pathway would be helpful**

We have now added further information to the referral pathway section which we believe makes the pathway clearer.

- **You state in the discussion that you use case studies and clinical vignette – but this isn’t clear in the methods. It focuses on the interview guide and a “hypothetical” case study – which doesn’t feel the same as saying you used case studies.**

Thank you for flagging this. We have now changed all mentions of the case stud/vignette to “hypothetical case study” for consistency.

11 Results

- **P8 starting line 42: I don’t really gain much insight into what behavioural marks or tasks were used.**

Thank you for this comment. We had added the following to description of the behavioural markers .

“Oft cited features of autism included atypical eye-contact, delayed language, fixed or specialised interests (e.g. US Emergency Departments), ritualistic behaviours (e.g. rocking), and sensory sensitives. When thinking about ADHD, most practitioners characterised the

condition by inattention, problems with concentration, impulsivity, social problems, and impaired academic functioning” P9

- **What professional networks were important?**

For clarity we have changed this to

“Discussions with colleagues” P9

- **You said it varied regarding triangulation, but what was the variation – did some not do it all? How was this judged?**

We have clarified the comment about triangulation

“Nevertheless, the extent to which GPs considered, used, and triangulated this information varied considerably, with some GPs offering to contact schools and others basing the referral on parental report. P9”

- **I think the results would also benefit by reflecting on NICE guidelines and how what was said links with this**

We have added a sentence on this:

“Of note, GPs did not refer to NICE guidance.”P9

- **Previous research has also mentioned about professionals feeling parent do not want a non-ADHD diagnosis (Study of user experience of an objective test (QbTest) to aid ADHD assessment and medication management: a multi-methods approach | SpringerLink)**

Thank you for flagging this interesting study. We really think this is such an interesting issue. However, given the word count coupled with the fact that this was not a strong theme in the results we do not feel we have the space to do that topic justice here.

- **Likewise other studies have looked at training GPs for ADHD awareness XXXXX**

Thank you for this comment. We have discussed Tatlow’s review which we regard as a fairly comprehensive evaluation of the literature on this topic.

12 Discussion

- **The Interviews were carried out pre-COVID, the authors should reflect on how services may have changed as a result of this and the relevance of their findings**

We completely agree and have added the following sentences to the limitations

“Finally, this research took place prior to Covid-19 pandemic. Therefore, with GPs under consider strain, it is important to consider whether how the pandemic might have shaped referral pathways and indeed GP’s attitudes towards identifying neurodevelopmental conditions.” P17

- **Other research has explored ways of improving GP knowledge of ADHD:**

<https://mededu.jmir.org/2020/2/e19871/>

Thank you for flagging this interesting study. We cite two systematic reviews on the topic and unfortunately we don’t feel like we have sufficient space to evaluate this study in detail.